# Neurolysin Knockout Mice in a Diet-Induced Obesity Model

**DOI:** 10.3390/ijms242015190

**Published:** 2023-10-14

**Authors:** Bruna Caprioli, Rosangela A. S. Eichler, Renée N. O. Silva, Luiz Felipe Martucci, Patricia Reckziegel, Emer S. Ferro

**Affiliations:** 1Pharmacology Department, Biomedical Sciences Institute (ICB), São Paulo 05508-000, SP, Brazil; brunacaprioli@usp.br (B.C.); eichler@usp.br (R.A.S.E.); oliveirarenee@gmail.com (R.N.O.S.); felipemartucci@gmail.com (L.F.M.); 2Department of Clinical and Toxicological Analysis, Faculty of Pharmaceutical Sciences (FCF), University of São Paulo (USP), São Paulo 05508-000, SP, Brazil; reckziegel.patricia@usp.br

**Keywords:** neurolysin, adipose tissue, neurotensin, intracellular peptides, obesity, diet-induced obesity, leptin, glucagon-like peptide-1

## Abstract

Neurolysin oligopeptidase (E.C.3.4.24.16; Nln), a member of the zinc metallopeptidase M3 family, was first identified in rat brain synaptic membranes hydrolyzing neurotensin at the Pro-Tyr peptide bond. The previous development of C57BL6/N mice with suppression of Nln gene expression (Nln^-/-^), demonstrated the biological relevance of this oligopeptidase for insulin signaling and glucose uptake. Here, several metabolic parameters were investigated in Nln^-/-^ and wild-type C57BL6/N animals (WT; n = 5–8), male and female, fed either a standard (SD) or a hypercaloric diet (HD), for seven weeks. Higher food intake and body mass gain was observed for Nln^-/-^ animals fed HD, compared to both male and female WT control animals fed HD. Leptin gene expression was higher in Nln^-/-^ male and female animals fed HD, compared to WT controls. Both WT and Nln^-/-^ females fed HD showed similar gene expression increase of dipeptidyl peptidase 4 (DPP4), a peptidase related to glucagon-like peptide-1 (GLP-1) metabolism. The present data suggest that Nln participates in the physiological mechanisms related to diet-induced obesity. Further studies will be necessary to better understand the molecular mechanism responsible for the higher body mass gain observed in Nln^-/-^ animals fed HD.

## 1. Introduction

Neurolysin (EC 3.4.24.16; Nln) belongs to the metallo-oligopeptidase family of proteases [1,2,3,4,5]. Nln was identified by immunohistochemistry in different intracellular compartments of rat brain neurons [6,7,8], and it can undergo alternative splicing to be addressed to the mitochondria [9]. The conformation of the Nln active site allows access only to peptides containing 5–17 amino acids, as its catalytic site is located at the base of a narrow and deep channel, which shapes its specificity for bioactive peptides without a developed secondary or tertiary structure [10,11,12]. Nln is capable of cleaving a series of neuropeptides [1,13,14,15] and intracellular peptides (InPeps) [16,17], and it has been suggested that it plays a biological function in pain control [18,19], and in blood pressure regulation [20,21]. Neurotensin, the first neuropeptide characterized as a substrate for Nln [2,3,22,23], is a small peptide with several functions: controlling food intake and energy balance, possibly through the regulation of gut lipid absorption and fat homeostasis [24]. Moreover, Nln has been shown to be encoded in two isoforms: a longer form containing a cleavable mitochondrial targeting sequence directed to mitochondria, and a shorter form that remains in the cytosol [9].

The generation of a C57BL6/N animal model of Nln gene knockout (Nln^-/-^) showed that its deletion is compatible with embryonic and adult life, and that Nln^-/-^ animals cannot be visually distinguishable from wild-type (WT) animals [16,17]. Between Nln^-/-^ and WT animals there was no distinction in muscle mass and fat mass, although Nln^-/-^ animals showed a slight, but significant, decrease in body mass compared to WT animals [16]. Findings from glucose and insulin tolerance tests indicated that Nln^-/-^ animals were more sensitive to insulin compared to WT animals [16]. Furthermore, Western blotting assays revealed increased levels of AKT phosphorylation in the gastrocnemius muscle and epididymal adipose tissue but not in the liver of Nln^-/-^ animals [16]. Moreover, Nln^-/-^ animals demonstrate a greater gluconeogenic capacity, with greater production of glucose from pyruvate [16]. Genes such as fructose 6-bisphosphatase and glucokinase that directly participate in the gluconeogenesis process in the liver showed greater expression in Nln^-/-^ animals compared to WT animals [16]. The differences observed in the metabolic phenotype of Nln^-/-^ animals emphasized the need for additional biological characterization, including a possible role in obesity.

Nln and thimet oligopeptidase (EC3.4.24.15; THOP1) are both members of the zinc metallopeptidase M3 family, containing a thermolysin-like catalytic domain [10,25]. Human Nln and THOP1 share 63% sequence identity over 677 common residues, and the crystal structures of both enzymes show that they adopt almost identical folds [10,11,12]. Both Nln and THOP1 are commonly distributed in different mammalian tissues, although with distinctive subcellular locations [6,8,26,27]. Both Nln and THOP1 have substrate size restriction, optimally metabolizing peptides containing from 8–13 amino acids involved in a range of physiological processes [5,6,7,25,28,29,30]. Both Nln and THOP1 have been shown to degrade several neuropeptides [1,19,26,31,32], and to metabolize peptides released by the proteasome [16,17,28,30]. Despite the biochemical similarities of Nln and THOP1 among most substrates, neurotensin (pyr-Glu-Leu-Tyr-Glu-Asn-Lys-Pro-Arg-Arg-Pro-Tyr-Ile-Leu-OH) is cleaved by Nln at the Pro^10^-Tyr^11^ bond, while THOP1 cleaves the Arg^8^-Arg^9^ bond [1]. Moreover, Nln and THOP1 can also be distinguished by their sensitivity to carboalkyls, phosphinic and Pro-Ile inhibitors [23,33,34], and their drastic, distinct sensitivity to thiol compounds [35,36,37,38]. 

Nln and THOP1 subcellular distribution in a rat brain was evaluated side-by-side using electron microscopic immunogold labeling [6]. In all areas of the rat brain examined, Nln and THOP1 immunoreactivity were observed in selective subpopulations of neuronal and glial cells. Subcellular localization of THOP1 in neurons revealed that this enzyme was predominantly concentrated in the nucleus, whereas Nln was almost exclusively cytoplasmic. Within the cytoplasm, both Nln and THOP1 were identified throughout the perikarya and dendrites, as well as within axons and axon terminals. However, whereas Nln was observed on both the cytoplasmic and luminal sides of cytoplasmic membranes, THOP1 was observed on the cytoplasmic face of the membranes. These results suggest that both THOP1 and Nln could play a major role in the metabolism of intracellular substrates, whereas Nln would primarily be involved in the processing and deactivation of extracellular signaling peptides [6]. Indeed, Nln soluble subcellular distribution was suggested to occur in glial cells, while cytoplasmic and membrane associated Nln occurs in neuronal cells [27]. Nln secreted from glial cells can diffusely metabolize neuropeptides in the extracellular milieu, while neuronal Nln can metabolize neuropeptides close to their receptor as a membrane-bound peptidase [8]. 

Obesity can be caused by multiple mechanisms, unbalancing the energy metabolism homeostasis [39,40]. Currently, there is an increased consumption of ultra-processed foods, generally rich in hydrogenated fats, simple carbohydrates and low levels of complex carbohydrates [41], as well as a decline in energy expenditure, which is associated with a lack of regular physical activity [42]. An excess of energy ingested and not expended accumulates as body fat, especially in white adipose (WAT) and hepatic tissues, representing one of the main reasons people are overweight and obese [39,43]. The WAT is known for its storing energy function in subcutaneous and visceral depots. Subcutaneous depot forms a layer in the hypodermis, while the visceral depot is subdivided into: (a) omental, adipose tissue that superficially surrounds the anterior region of the intestine; (b) mesenteric, deeper adipose tissue, which surrounds the posterior region of the intestine close to the spine; (c) retroperitoneal, located near the kidneys, in the dorsal region of the abdominal cavity; (d) gonadal, located around the gonads and gonadal ducts. In addition to these two main depots (subcutaneous and visceral), there are small depots of visceral adipose tissue in the heart (epicardial adipose tissue), stomach (epigastric adipose tissue) and blood vessels (perivascular adipose tissue) [44]. 

High-fat diets are commonly used to induce obesity in animals, since they generate adverse metabolic effects, which includes obesity [45]. THOP1 knockout C57BL6/N mice (THOP1^-/-^) were fed for 24 weeks with either a standard diet (SD; caloric content of 3.8 kcal/g; 70% carbohydrate, 20% protein, 10% fat) or a high-fat diet (HD; caloric content of 5.4 kcal/g; 25.9% carbohydrate, 14.9% protein, 59% fat) [28]. THOP1^-/-^ mice gained 75% less body weight and showed neither insulin resistance nor non-alcoholic fatty liver steatosis when compared to WT mice [28]. The expression of a phenotype emerges through intricate macromolecular interactions [46,47,48,49,50,51,52]. Within cells, complex signaling pathways orchestrate and regulate these interactions, and alterations in these have been used to predict new targets for treatments of human diseases [46,47,48,50,53,54,55]. InPeps generated by the orchestrated intracellular action of the proteasome and peptidases such as Nln and THOP1 [56,57,58,59,60,61], may be physiologically functional modulating the G protein-coupled receptors signal transduction [62,63,64,65,66], altering protein–protein interactions [67,68] and interfering with gene expression, possibly through physical interaction with specific microRNAs [28]. 

Here, Nln^-/-^ was challenged by a high-fat/high-sugar diet-induced obesity model [69], to advance the understanding of the possible biological relevance of Nln in energy metabolism and obesity. The combination of high-fat/high-sugar in a diet-induced obesity model was shown to induce a pronounced obesity after feeding the animals for up to eight weeks [69,70]. Body mass and several other metabolic parameters were evaluated in both male and female animals, comparing Nln^-/-^ to WT animals fed either SD or HD.

## 2. Results

### 2.1. Body Mass, Food and Water Consumption

WT or Nln^-/-^ animals were weighed before starting the SD or HD (Week “zero”, Figure 1A,B). Importantly, none of these mice showed statistically significant differences in their initial body mass (Figure 1A,B). The individual body mass of each of these animals—WT and Nln^-/-^, male and female, fed either SD or HD—were checked once a week during the following seven weeks (Figure 1A,B).

After seven weeks fed HD, all animals had a greater body mass compared to those fed SD, regardless of genotype or gender (Figure 1A,B). Nln^-/-^ animals, both male and female, showed a significantly higher body mass after only four weeks on HD, compared to their respective control WT animals fed HD (Figure 1). The total body mass gain of male WT animals fed SD was 2.67 g ± 0.49 g, while the mass gain of Nln^-/-^ male animals fed SD was 4.0 g ± 0.55 g. After seven weeks, male WT animals fed HD gained 8.87 g ± 3.6 g of body mass, while male Nln^-/-^ animals, also fed HD, gained 14.0 g ± 1.29 g of body mass. WT females fed SD gained 4.28 g ± 0.28 g of body mass in relation to their initial mass, while WT females fed HD gained 5.8 g ± 0.37 g of body mass compared to their initial mass. SD-fed Nln^-/-^ females gained a body mass of 3.83 g ± 0.6 g, while Nln^-/-^ females fed HD gained 10.88 g ± 10.2 g of body mass. Therefore, after seven weeks, all animals fed HD gained significantly more body mass than their respective controls fed SD (Figure 1C,D). Nonetheless, after being fed a HD for seven weeks, the body mass gain of the Nln^-/-^ animals, regardless of gender, was significantly higher compared to their respective WT controls fed HD (Figure 1C,D).

Both WT and Nln^-/-^ mice, male and female, fed SD consumed more food than their control animals fed HD (Table 1 and Table 2). There was no variation in water and calorie consumption between male groups (Table 1); however, water consumption was slightly higher in female fed HD (Table 2).

Comparing the wet mass of individual tissues (liver, inguinal adipose tissue, gonadal adipose tissue, retroperitoneal adipose tissue and gastrocnemius), among male and female animals fed with SD, regardless of genotype or gender, no statistically significant differences were observed (Table 3 and Table 4). Liver tissue wet mass was reduced in animals fed HD compared to those fed SD, regardless of genotype or gender (Table 3 and Table 4). An increase was observed in the wet mass of all adipose tissues evaluated (inguinal, gonadal and retroperitoneal) from animals fed HD, compared to those fed SD, regardless of genotype or gender (Table 3 and Table 4). Nln^-/-^ female animals have a greater increase in all adipose tissue depots investigated herein, compared to their respective WT control animals (Table 4). However, Nln^-/-^ male animals only showed a significant increase in the wet mass of retroperitoneal adipose tissue depot, compared to their respective WT control animals (Table 4). These data suggest the distinctive metabolic behavior of Nln^-/-^ inguinal and gonadal adipose tissues in male and female animals. Moreover, male animals, Nln^-/-^ or WT, fed HD, had a lower weight of gastrocnemius muscle (Table 3). Female Nln^-/-^ animals fed with HD showed a lower wet weight of the gastrocnemius muscle, compared to the respective Nln^-/-^ fed SD (Table 4).

WT and Nln^-/-^ animals, male and female, fed with SD or HD for seven weeks, were submitted to GTT (Figure 2). WT and Nln^-/-^ male animals HD-fed presented lower glucose uptake, compared to their respective controls fed SD (Figure 2A,C). Similar results were not observed for both WT and Nln^-/-^ female animals; although Nln^-/-^ female animals fed HD presented higher glucose uptake compared to WT female fed HD (Figure 2B,D).

Blood insulin measurement suggested that only Nln^-/-^ females fed HD, have increased plasma insulin concentration (Figure 3). These results could corroborate the higher glucose uptake observed for Nln^-/-^ female animals fed HD, compared to WT female fed HD (Figure 2B,D).

ITT was performed on WT and Nln^-/-^, male and female animals, fed either SD or HD for seven weeks, with blood glucose monitored at 4 min intervals for 16 min (Figure 4). Despite the absence of statistical significance, Nln^-/-^ male animals fed SD seem to have a better glucose uptake and higher insulin sensitivity (Figure 2 and Figure 4A,C). Male Nln^-/-^ animals fed HD showed a statistically significant reduction in insulin sensitivity (Figure 4C); these data suggest that HD reverses the higher insulin sensitivity of Nln^-/-^ male animals. No differences were observed among female animals fed either SD or HD, regardless of their genotype.

### 2.2. Analysis of Gene Expression

Gene expression analyses of adipose tissues from WT and Nln^-/-^ animals, male (Table 5 and Table 6) and female (Table 7 and Table 8), were performed herein. Possible variation on specific gene expression was defined comparing to the expression levels of constitutively expressed RPL19 or cyclophilin A mRNA. Analyzing gene expression on the adipose tissue of mice fed a HD-induced obesity, both RPL19 and cyclophilin A have been suggested the most appropriate reference housekeeping genes [71].

Both WT and Nln^-/-^ animals, male and female, fed a HD presented increased or decreased expression of several genes related to energy metabolism, compared to control animals fed SD (Table 5, Table 6, Table 7 and Table 8). The effect of the Nln^-/-^ phenotype could be observed in both male and female animals, fed either SD or HD (Table 5, Table 6, Table 7 and Table 8). Increased expression of leptin was commonly observed in male and female Nln^-/-^ animals fed with HD, compared to WT animals on the same HD diet (Table 5, Table 6, Table 7 and Table 8). Among several additional differences, increased gene expression of PPAR-alpha was observed comparing animals fed SD *vs* HD, and also between Nln^-/-^ and WT females fed SD (Table 5, Table 6, Table 7 and Table 8). In addition, increased gene expression of LPL, IDE, ACE1 and ß5-Prot was observed in male animals fed HD *vs* SD; for ACE1 the difference was only seen with RPL19 as the housekeeping reference gene (Table 5 and Table 6). Reduced gene expression in SD-fed Nln^-/-^ male animals, compared to WT male animals also fed SD, was observed for PPAR gamma and ß3-adrenoceptor (ADBR3) (Table 5 and Table 6). A reduction in the expression of PGC-1 alpha was observed in Nln^-/-^ SD-fed female animals, compared to WT female animals SD-fed (Table 7 and Table 8).

Increased gene expression in Nln^-/-^ male animals fed a HD, compared to WT male animals also fed a HD, was observed for leptin and ß5-Prot (Table 5 and Table 6). Reduced gene expression in HD-fed Nln^-/-^ male animals, compared to WT male animals that were also fed HD, was observed for PPAR gamma and ADBR3; these were the same genes downregulated in SD-fed Nln^-/-^ animals (Table 5 and Table 6). In Nln^-/-^ female animals HD-fed, an increase in the expression of FAS and leptin was observed, compared to WT female animals that were also fed HD (Table 7 and Table 8). In HD-fed Nln^-/-^ female animals, a reduction in the expression of ADBR3 was observed, compared to WT female animals that were also fed HD (Table 7 and Table 8). Therefore, similar results were largely observed using either RPL19 or cyclophilin A as the reference housekeeping gene (Table 5, Table 6, Table 7 and Table 8). The variation observed in the expression levels could be related to cyclophilin/RPL19 ratios. These results are in accordance with previous evidence, suggesting that both *RPL19* and cyclophilin could be used as housekeeping genes for sex-unbiased human and mouse gene expression profile [71,72].

## 3. Discussion

The major finding presented here was that male and female Nln^-/-^ animals fed a HD, have both higher food intake and body mass gain compared to WT animals HD-fed. A higher increase in the gene expression of leptin was observed in male and female Nln^-/-^ animals fed a HD, compared to WT animals HD-fed. Secretion of leptin is well known to regulate the appetite and food intake, which helps to control body weight [73]. In obesity, despite an increment in leptin expression and secretion, the control of food intake and appetite by leptin is largely reduced due to leptin resistance, which seems related to alteration of class I cytokine leptin-receptor mediated signal transduction and leptin transport across the blood-brain barrier [74]. Crosstalk between cytokine receptors and G-protein coupled receptors, including to the beta-2 adrenergic receptor, have been previously described [75,76]. InPeps have been previously shown to modulate signal transduction of G-protein coupled adrenergic receptors [62,64]. However, further experiments are necessary to characterize the possible involvement, and the molecular mechanisms of action, of InPeps and neuropeptides, substrates or products of Nln, in the leptin resistance of animals fed a HD. These data suggest the physiological participation of Nln in energy homeostasis in a model of diet-induced obesity.

Despite their genotype or gender, animals fed HD gained more body mass while ingesting smaller amounts of food, compared to animals that were fed SD; similar results were observed in a previous study using a similar diet-induced obesity model with sweetened condensed milk and multivitamin complex addition [69]. The relatively short period of time these animals were fed either with SD or HD may have impacted the low body mass gain of WT animals HD-fed, compared to SD-fed WT animals. The WT mouse strain employed herein may also have impacted the low body mass increment observed, as previously reported [77,78,79]. It has been previously shown that substrate utilization and thermogenesis significantly change following ingestion of different types of carbohydrates in young healthy lean male volunteers [80]. Therefore, the carbohydrate composition of the diet should be expected to have critical outcomes for energy and macronutrient balance in rodents as well. Indeed, distinct proteins, carbohydrates and lipids have different metabolic roles in energy homeostasis, and because of that, it has been suggested that diets of similar overall energy content, but with different macronutrient distribution, distinctively modify the appetite, metabolism and thermogenesis [81]. Ingestion of sweetened condensed milk that contains 53% sucrose, and causes a rapid increase in plasm insulin levels, induces energy storage as fat in the adipose tissue, which is directly associated with the development of obesity [82,83]. Moreover, micronutrients may also exacerbate obesity-related metabolic consequences, impairing insulin signaling and dysregulating glycemic control; vitamins A, B1, B6 and B12, as well as selenium and zinc, have been reported to be low in the plasma of obese individuals [84,85]. Nln^-/-^ mice were shown herein to be even more sensitive than WT mice to obesity induced by a HD; Nln^-/-^ male and female animals fed a HD ingested more food and gained more body mass than WT animals fed HD. That could occur through distinctive InPeps and neuropeptides, both in central and peripherical tissues, that differentially modulate the complex macromolecular interaction network responsible for gene expression and protein–protein interactions controlling energy balance and storage [47,48,49,50,73]. In fact, the peptidome of Nln^-/-^ mice also have been previously determined, showing alteration in the relative concentration of InPeps in soleus and gastrocnemius muscles, liver, and epididymal adipose tissue [16], and also of neuropeptides such as Met-enkephalin and octapeptide in the brain [17]. Previously, THOP1^-/-^ male and female mice were shown to gain less body weight than WT control mice, both fed a similar HD [28]. Specific InPeps identified in the adipose tissue of HD-fed THOP1^-/-^ mice were suggested to participate in the observed obesity-resistance phenotype [28]. The molecular mechanism was suggested to involve the physical interaction between InPeps and specific microRNAs, which would lead to an alteration in controlling the expression of genes involved in energy metabolism [28]. Therefore, a differential profile of InPeps and neuropeptides in Nln^-/-^ could be driven its distinctive metabolic routes during the diet-induced obesity model used herein, which should be relevant to the different phenotype of Nln^-/-^ animals characterized herein.

Leptin, the product of the obese (*ob*) gene [86], is a key adipocyte-secreted hormone for energy balance, and is involved in obesity development [87,88,89]. Leptin circulates in blood and acts on the brain to regulate food intake and energy expenditure [89]. The body weight-reducing effects of leptin were not seen augmented by enhanced thermogenesis, suggesting that leptin has effects on body temperature regulation, by opposing torpor bouts and by shifting thermoregulatory thresholds; the central pathways behind these effects are largely unexplored [90]. Leptin-deficient ob/ob mice have fully functional brown adipose tissue, and leptin treatment could not increase thermogenesis both in WT and leptin ob/ob mice [91]. Therefore, leptin in parallel to the effects of other cytokines alters centrally regulated thresholds of thermoregulatory mechanisms [91]. Future experiments investigating Nln^-/-^ and WT animals, fed either SD or HD, should shed light on the possible contribution of Nln and InPeps to thermoregulation [66,92,93,94,95,96].

Gene expression can be correlated with increased confidence to protein expression [97], and strong linear correlations were previously observed between plasma leptin, leptin secretion, and leptin mRNA levels [98]. Therefore, the relative abundance of several mRNAs was evaluated herein using both RPL19 and cyclophilin A as housekeeping genes, to gain molecular insights into the diet-induced obese phenotype observed for Nln^-/-^ animals [99]. The higher increase in the leptin hormone gene expression could be one of the most relevant for the obese phenotype of Nln^-/-^ animals. Thus, considering the RPL19 housekeeping gene as a reference, while leptin gene expression increased approximately sixteen times in male WT animals fed HD, it increased approximately forty-six times in male Nln^-/-^ animals, compared to their respective control animals fed SD. In WT female animals fed HD leptin gene expression increased eight times, while in female Nln^-/-^ animals fed HD leptin gene expression increased approximately twenty-five times compared to control animals fed SD, considering the RPL19 housekeeping gene as a reference. Leptin levels increase when fat mass increases to suppress appetite until weight is lost [89]. However, leptin resistance can occur to increase the predisposition of individuals to diet-induced obesity, which in turn contributes to a further increase in leptin levels and aggravation of existing leptin resistance in a vicious cycle [100]. Several mechanisms are related to leptin resistance, including (1) structural changes to the leptin hormone itself, (2) its transport across the blood-brain barrier, (3) the misfunctioning of the leptin-receptor and signal transduction through the JAK–STAT signaling pathway [74]. Neurotensin, which was the first neuropeptide characterized as a substrate for Nln [2,3,22,23], has several functions, including the control of food intake and energy balance, possibly through the regulation of gut lipid absorption and fat homeostasis [24]. Neurotensin is also one of the peptides that mediate the hypothalamic action of leptin on feeding [101,102]. Unbalanced neurotensin metabolism in Nln^-/-^ fed HD could significatively impact the leptin-sensitive neural circuitry, causing leptin resistance, considering that neurotensin is a well-known substrate of Nln [1,103]. Further investigations need to be conducted to determine possible changes in hypothalamic neurotensin levels, both in WT and Nln^-/-^ animals fed either SD or HD. In addition, leptin resistance in Nln^-/-^ and WT animals fed HD under the present diet-induced obesity model, with sweetened condensed milk and multivitamin complex addition, also need to be investigated.

The expression of additional genes related to energy metabolism was also modified. That includes a reduction in ADBR3 expression observed in both Nln^-/-^ male and female animals HD-fed, compared to WT animals also fed a HD. The decrease in ADBR3 levels could generate an increase in lipid content, due to a decrease in lipolysis mediated by adrenergic receptor stimulation. Thus, the gain in adipose tissue mass observed in HD-fed Nln^-/-^ animals could also involve reduced lipolysis. The adipose tissue expression of ADBR3 was higher in THOP1^-/-^ male animals fed a HD, which distinctively from Nln^-/-^ were resistant to diet-induced obesity [28]. Interestingly, the involvement of THOP1 metabolizing InPeps that modulates adrenergic signaling transduction has been well documented [62,64]. These data suggest that Nln and THOP1 could be involved in the metabolism of InPeps within adipose cells, modulating the adrenergic signaling transduction and energy balance.

The gene expression of several endopeptidases (NEP, POP, DPP4, IDE and ECA1), and that of the proteasome ß5-Prot subunit, was evaluated herein. Despite NEP and POP, modulation of DPP4, IDE, ECA1 and ß5-Prot subunit expression were observed herein. In Nln^-/-^ male animals fed SD, an increased expression of IDE, ECA1 and ß5-Prot subunit was observed. These data suggesting that, in the absence of Nln, a compensatory expression of IDE, ECA1 and ß5-Prot subunit expression occurred. In parallel, the expression of IDE and proteasome ß5-Prot subunit was observed both in WT and Nln^-/-^ male animals fed HD, considering the relative expression to both RPL19 and cyclophilin A. Interestingly, previous data suggest that THOP1 expression was not altered in Nln^-/-^ animals, and *vice–versa* [16,17,28,29]. IDE predominantly metabolize peptides inside the cells [104], similarly to Nln [6,8,26,27] and THOP1 [25,65], corroborating previous suggestions that the intracellular peptide metabolism is modulated under a diet-induced obesity model [14,25,28,29,30,63,65,105,106]. DPP4 and ACE1 gene expression increased WT and Nln^-/-^ female animals fed HD, considering both RPL19 and cyclophilin A as housekeeping genes. Both DPP4 and ACE1 metabolize peptides on the extracellular milieu [107,108], suggesting that in females, conversely to males, the extracellular peptide metabolism plays a major role in diet-induced obesity. Interestingly, DPP4 preferentially degrades peptides containing an alanine or proline at the penultimate position (P1 position), which includes the glucagon-like peptide-1 (GLP-1) [109,110]. Physiologically, GLP-1 is known to control body weight gain inhibiting appetite and delaying gastric emptying, and GLP-1 synthetic orally-active receptor agonist semaglutide has now been used to control type-2 diabetes and obesity [111,112,113,114]. DPP4 inhibition has shown protective effects against type 2 diabetes and several metabolic disorders, including obesity [109]. Therefore, an increased DPP4 gene expression directly correlates with the increased body weight gain of WT and Nln^-/-^ females fed HD presented here. These data corroborate the significant role of intracellular and extracellular peptide metabolism in diet-induced obesity. The extent to which the mitochondrial absence of Nln plays a relevant role in the metabolic alterations of these animals needs to be investigated [9]. Gender differences on peptide metabolism also deserve further attention, particularly in a diet-induced obesity condition.

Previous studies on the characterization of Nln^-/-^ fed SD showed a greater sensitivity to insulin, shown by increased AKT phosphorylation in muscle and adipose tissue but not in the liver [16]. Here, male Nln^-/-^ animals fed HD showed a statistically significant reduction in insulin sensitivity compared to Nln^-/-^ animals fed SD. Moreover, Nln^-/-^ female animals HD-fed showed high insulin levels compared to WT female animals fed either SD or HD, whereas no alteration of this parameter was observed for male animals, regardless of their genotype and diet. These data were in agreement with enhanced blood glucose uptake of HD-fed Nln^-/-^ female animals, compared to HD-fed WT animals. Metabolic differences between males and females have been reported, demonstrating that female mice express higher contents of the glucose transporter (GLUT4) in white adipose tissues than males [115]. Glycemic regulation occurs through several mechanisms, including the participation of the GLUT4 transporter and insulin secretion, whose deficiency can generate insulin resistance evolving to type 2 diabetes [116,117,118]. Further experiments are necessary to investigate the molecular mechanism related to the lack of higher insulin sensitivity of Nln^-/-^ male animals fed HD, which could be related to the alteration of insulin receptor signal transduction pathways.

The distinctive peptide profiles of WT and Nln^-/-^ in the brain, adipose and liver tissues have been previously shown, suggesting that Nln can metabolize bioactive peptides both inside and outside the central nervous system [16,17]. One of the peptides identified in the liver of Nln^-/-^ was named Ric4 (LASVSTVLTSKYR), and was shown to decrease blood glycemia, following either intraperitoneally or oral administration to WT mice [119]. Therefore, it was suggested that Ric4 contributes to the enhanced glucose uptake and insulin sensitivity observed in Nln^-/-^ [119]. Pep19, identified by means of conformational-sensitive antibodies targeting G-protein coupled receptors [120,121], induced uncoupling protein 1 expression in both white adipose tissue and 3T3-L1 differentiated adipocytes [96] Oral administration of Pep19 into diet-induced obese Wistar rats or Swiss mice significantly reduces whole body weight and the size of adipocytes [95,96]. THOP1 was also shown to play a key role in the metabolism of peptides regulating energy expenditure [28]. However, while Nln^-/-^ mice were shown herein to be more susceptible to diet-induced obesity, THOP1^-/-^ mice were resistant to diet-induced obesity after being fed a HD for 24 weeks [28]. It has been suggested that the distinctive subcellular localization of Nln and THOP1 drives their biological significance [6], which, together with their biochemical specificities, could be of importance to the phenotype differences observed in diet-induced obesity. Taken altogether, these data corroborate the suggestion that both oligopeptidases THOP1 and Nln play key roles in regulating energy metabolism, possibly through InPeps and neuropeptide metabolism. Further investigations are necessary to understand the molecular basis of the distinctive diet-induced obesity phenotype of Nln^-/-^ and THOP1^-/-^ animals.

In conclusion, the present report suggests that Nln plays a key physiological function in regulating energy metabolism.

## 4. Materials and Methods

### 4.1. Animals

Male and female WT C57BL6/N mice, eight-weeks-old, were obtained from the central animal facility of the University of São Paulo Medical School, São Paulo, SP, Brazil. Male and female Nln^-/-^ mice (on the C57BL6/N genetic background), eight-weeks-old, were obtained from the local animal facility from the Pharmacology Department, Biomedical Science Institute, University of São Paulo, São Paulo, SP, Brazil. Both animal facilities have a certificate of quality in biosafety granted by the Brazilian National Technical Commission of Biosafety from the Ministry of Science, Technology and Innovation. A full detailed description of Nln^-/-^ mice generation and genotyping procedures was previously described [16,17]. All animals lived and shared the same environment, and the same persons maintained them during the entire experimental process. The mice were maintained in individually ventilated cages (Ventilife, ALESCO, São Paulo, SP, Brazil), in groups of 3–5 animals *per* individually ventilated cages, at 22 °C, under standardized conditions, with an artificial 12 h dark–light cycle, with *ad libitum* access to chow and drinking water. The experiments started with eight-week-old animals and were conducted for an additional seven weeks. 

All experimental procedures involving animals were previously approved by the Committee on Ethics in the Use of Animals of the Biomedical Sciences Institute, University of São Paulo, São Paulo, SP, Brazil, under number 4112010621, in accordance with the precepts of Law 11,794 of 8 October 2008, with Decree 6899 of 15 July 2009, as well as with the rules issued by the Brazilian National Council for the Control of Animal Experimentation (CONCEA, Brazil). Animals were maintained at the animal facility of the Department of Pharmacology, located in the Biomedical Sciences I, University of São Paulo, Capital. 

### 4.2. Mice Genotyping

The animals used for the experiments had their genotypes analyzed and confirmed using conventional polymerase chain reaction (PCR), as previously described [16]. DNA samples were extracted from mice tails (3 mm), which were digested overnight at 55 ºC with 50 µL of 100 mM Tris-buffer, pH 7.4, containing 5 mM EDTA, 200 mM NaCl, 0.2% SDS containing and 80 µg of proteinase K (Sigma, São Paulo, Brazil). The PCR genotyping assays used the following oligonucleotide sequences: pNlnF3 (5′ CGCCTCCTGCACCTACCA 3′); pNlnwtR3 (5′ ATTTGCCAGGTTAAGAGATCG 3′), which generated fragments of 629 bp for the WT mice; and the pNlnkoR2 (5′ CGTGTCCTACAACACACACTCC 3′), which generated a 312 bp fragment for the Nln^-/-^ mice. Briefly, 2 µL of DNA extracts (1:10), 3 µL 10x PCR Buffer (100 nM Tris-HCl, pH 8.3; 500 mM KCl, 15 mM MgCl2; 0, 01%), 1 µL 50 mM MgCl2, 1 µL of 5 mM dNTP, 0.5 µL of each primer (10 µM) and 0.2 µL of 5 U/µL TaqDNA polymerase (all PCR reagents were from Thermo Fisher, São Paulo, SP, Brazil), were added for a 30 µL of PCR reaction solution. The reactions were conducted in a conventional thermal cycler under the following conditions: 95 °C for 5 min, 40 cycles of 95 °C for 30 s, 60 °C for 30 s, 72 °C for 30 s, and the reaction was completed after incubation at 72 °C for 5 min. The reaction products were submitted to electrophoresis in a 2% agarose gel, containing the dye Syber Safe (Thermo Fisher, São Paulo, SP, Brazil), visualized and photographed in the ChemiDoc MP Imaging System (BioRad, Hercules, CA, USA). Mice genotyping shows the presence of a 629pb band in WT animals, which was absent in Nln^-/-^ animals. Conversely, a 312pb band was present in Nln^-/-^ animals and absent in WT animals.

### 4.3. Diet-Induced Obesity

The diet-induced obesity model used herein was previously described [69]. Briefly, groups of eight-weeks old WT and Nln^-/-^ animals (n = 5–8) were offered ad libitum either standard chow (SD) or high-fat diet chow supplemented with sweetened condensed milk [70] containing AIN-93 multivitamin complex (HD) [69]. The SD composition was carbohydrate, 70%; protein, 20%; fat, 10% (3.8 Kcal/g; Nuvilab CR1, Nuvital Nutrientes S.A., Colombo, PR, Brazil). The HD composition was carbohydrate, 27.44%; protein, 13.55%; fat, 59% (5.3 Kcal/g; Rhoster, Araçoiaba da Serra, SP, Brazil); supplemented ad libitum with sweetened condensed milk (53% sucrose, 15% lactose, 23% fat, 9% protein; 3.25 Kcal/g; Moça, Nestlé, São Paulo, SP, Brazil), and 1% of AIN-93 multivitamin and minerals complex (vitamins: A, 4.00 IU/g; D3 (added), 1.00 IU/g; E, 78.80 IU/g; K (as menadione), 0.75 ppm; thiamine hydrochloride, 6.00 ppm; riboflavin, 6.50 ppm; niacin, 30.00 ppm, pantothenic acid, 16.00 ppm; folic acid, 2.10 ppm; pyridoxine, 5.80 ppm; biotin, 0.20 ppm; B12, 28.00 mcg/kg; choline chloride, 1,250.00 ppm. minerals: calcium, 0.50%; phosphorus, 0.31%; potassium, 0.36%; magnesium, 0.05%; sodium, 0.13%; chlorine 0.20%; fluorine, 1.00 ppm; iron, 39.00 ppm; zinc, 35.00 ppm; manganese, 11.00 ppm; Copper, 6.00 ppm; iodine, 0.21 ppm; chromium, 1.00 ppm; molybdenum, 0.14 ppm; selenium, 0.22 ppm; Rhoster, Araçoiaba da Serra, SP, Brazil).

### 4.4. Measuring Animals Body Mass, Food, and Water Consumption

Weekly, WT and Nln^-/-^ animals were individually weighed on a conventional digital scale. The body mass week gain from male or female mice, WT or Nln^-/-^, fed SD or HD was determined. After the seven weeks on either SD or HD, the final body mass of the animals (male or female, WT or Nln^-/-^) was subtracted from the initial body mass to estimate the total body mass gain along the seven weeks of the experiments. The consumption of SD or HD was measured *per* ventilated cages (3–5 animals were housed *per* ventilated cage), using a conventional digital scale. The calculated chow (food) and/or sweetened condensed milk consumption of each mouse was estimated by dividing the total amount consumed per ventilated cage, by the number of animals that were housed in that specific cage (from 3–5 animals *per* ventilated cage). Every week the unconsumable SD or HD remaining in the cages was weighed using a conventional digital scale, and the remaining weight value was subtracted from the initial amount offered. The volume of water ingested was monitored using an appropriate conical graduated tube. Every week the volume of undrinked water was measured using an appropriate conical graduated tube, and the remaining volume value was subtracted from the initial volume offered. The above measurements allowed to closely estimate the amount of SD, HD and water ingested. The total calories consumed were calculated by multiplying 3.8 Kcal/g of ingested food for the animals fed SD, or by multiplying 8.55 Kcal/g of ingested food (corresponding to the sum of 5.3 Kcal/g of chow + 3.25 Kcal/g of the sweetened condensed milk) for the HD animals. At the end of seven weeks, animals were anesthetized with isoflurane and euthanized by cardiac puncture. Tissue samples from the liver, inguinal, retroperitoneal and gonadal adipose tissues, as well as the gastrocnemius muscle, were collected, weighed and stored in a freezer at −80°C until use. The tissue wet mass value was corrected by the total body mass of the animals, using the following equation: tissue mass × 100/body mass of the live animal.

### 4.5. Glucose Tolerance Test (GTT) and Insulin Tolerance Test (ITT)

Mice were previously submitted to fasting for 12 or 4 h for GTT or ITT measurements, respectively. For GTT, blood samples were collected from the tail at 0, 15, 30, 45, 60 and 90 min, after the intraperitoneal (*i.p.*) administration of a glucose 2 g/kg solution in Milli-Q water. For TTI, after the injection of 0.75U/kg of insulin blood samples from the tail were collected at 0, 4, 8 12 and 16 min. For both GTT and ITT, blood glucose was measured using a glucose meter Accu-Check Performa (Roche, São Paulo, SP, Brazil).

### 4.6. Plasma Insulin Measurements

Plasma insulin was determined in blood samples from WT and Nln^-/-^ animals, collected by cardiac puncture. The quantitative evaluation of insulin was performed in plasma by enzyme-linked immunosorbent assay (ELISA), using the insulin dosage kit for mice, according to the manufacturer’s instructions (EMD Millipore, St. Louis, MO, USA).

### 4.7. Real-Time PCR (qRT-PCR)

Real-time PCR (qRT-PCR) experiments, as previously described [28], were performed from total RNA extracted from retroperitoneal adipose tissue, to determine the expression levels of distinctive mRNAs (Table 9): peroxisome proliferator activated receptor gamma (PPAR-gamma), peroxisome proliferator activated receptor alpha (PPAR-alpha), fatty acid synthase (FAS), lipoprotein lipase (LPL), leptin, CD36 molecule (CD36), peroxisome proliferator-activated receptor gamma coactivator 1-alpha (PGC 1 alpha), mannose receptor (CD206), integrin alpha X (CD11C), adhesion G protein-coupled receptor E1 (F4/80), fatty acid-binding protein 4 (F4), adrenergic receptor beta 3 (ADBR3), neprilysin endopeptidase (NEP), angiotensin I converting enzyme (ACE1), prolyl oligopeptidase (POP), insulin degrading enzyme (IDE), dipeptidyl peptidase 4 (DPP4), proteasome subunit beta 5 (Prot-β5). Tissues were quickly removed, frozen in liquid nitrogen and stored in a −80°C freezer until use. Samples were homogenized, and total RNA was extracted using Trizol (Trizol^®^ Products Life Technologies, São Paulo, SP, Brazil). Total RNA integrity was verified by 1% agarose gel electrophoresis. cDNAs were synthesized from 2 µg of total RNA, using the cDNA High-Capacity kit (Thermofisher, Waltham, MA USA). Sample dilution curves were performed for assay standardization, in order to determine the efficiency of amplification of messenger RNAs (mRNAs) of target genes. The quantitative PCR assay was performed using Sybr Green Master Mix (Applied Biosystems, Waltham, MA, USA) with 100 nM of primer and 20 ng of cDNA and the runs performed in the QuantStudio 3 equipment (Applied Biosystem, Thermofisher, Waltham, MA, USA). The expression of mRNAs of interest was normalized by the expression of the reference mRNA and expressed as relative values using the “2ddCt” method, as previously described [28]. Both RPL19 and cyclophilin (Table 9) were used as housekeeping genes for internal relative controls, as previously suggested [71]. Target gene expression levels of WT animals were compared to Nln^-/-^, fed either SD or HD.

### 4.8. Statistical Analyses

The data obtained were analyzed for statistical significance using the GraphPad Prisma 6 software (GraphPad Inc.; San Diego, CA, USA), and were presented as mean ± standard error of the mean (SEM). Differences were determined by a non-parametric *t*-test for independent samples and the two-way ANOVA test for statistical analyses of 2 or more groups followed by the Holm–Sidak post-hoc test. *p* values < 0.05 were considered statistically significant for all analyses.

## Figures and Tables

**Figure 1 ijms-24-15190-f001:**
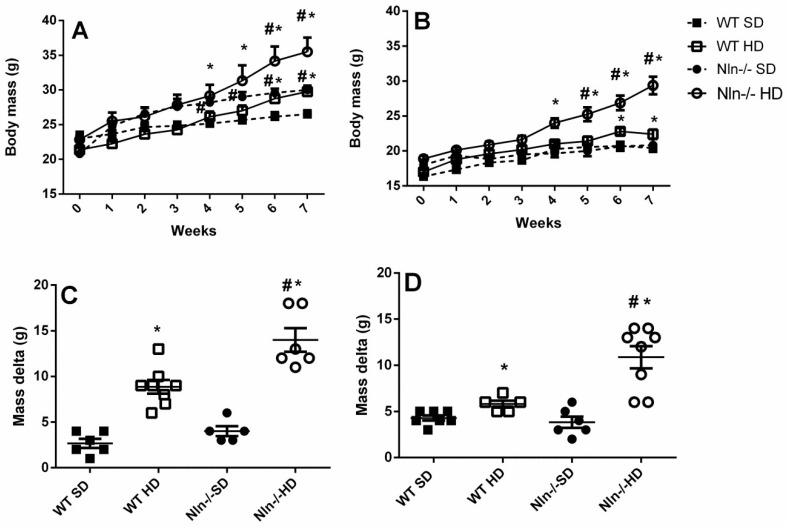
Weight gain of animals over seven weeks. A and B: body mass week gain in male (**A**) or female (**B**) mice, WT or Nln^-/-^, fed SD or HD diet. C and D: the final body mass was subtracted from the initial body mass (body mass gain) of male (**C**) or female (**D**) mice, WT or Nln^-/-^, fed SD or HD. (A–D): *, *p* < 0.05 between WT SD vs. WT HD or Nln^-/-^ SD vs. Nln^-/-^ HD; ^#^, *p* < 0.05 between WT SD vs. Nln^-/-^ SD or WT HD vs. Nln^-/-^ HD. Statistical analyses were performed using two-way ANOVA with the post-hoc Holm–Sidak test. Data are presented as mean ± SEM. n = 5–8 animals per group. Each individual group (WT, SD; WT, HD; Nln^-/-^, SD; Nln^-/-^, HD) was indicated with a specific symbol shown on the figure legend.

**Figure 2 ijms-24-15190-f002:**
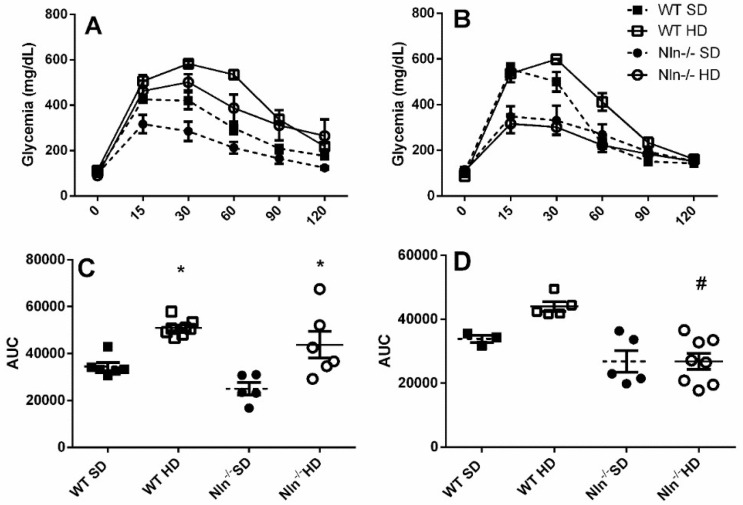
Glucose tolerance test (GTT) in male (**A**,**C**) or female (**B**,**D**) animals. (**A**,**B**), glycemic curves over the 120 min interval, (**C**,**D**), the area under the glycemic curves of the WT or Nln^-/-^ animals fed SD or HD. Intraperitoneal administration of glucose (2 g/Kg) was followed by blood glucose measurement with AccuCheck glycostrip (Roche, São Paulo, Brazil), at 15, 30, 60, 90 and 120 min. A sample collected and evaluated before glucose administration was considered as fasting blood glucose (time 0). A–D: *, *p* < 0.05 between WT SD vs. WT HD or Nln^-/-^ SD vs. Nln^-/-^ HD; ^#^, *p* < 0.05 between WT SD vs. Nln^-/-^ SD or WT HD vs. Nln^-/-^ HD. Statistical analyses were performed using two-way ANOVA with the post-hoc Holm–Sidak test. Data are presented as mean ± SEM. n = 5–8 animals per group.

**Figure 3 ijms-24-15190-f003:**
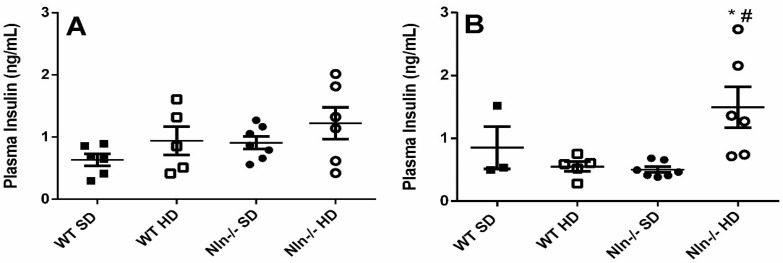
Quantitative assessment of plasma insulin by ELISA. (**A**,**B**): respectively, male and female animals. **A** and **B**: *, *p* < 0.05 between WT SD vs. WT HD or Nln^-/-^ SD vs. Nln^-/-^ HD; ^#^, *p* < 0.05 between WT SD vs. Nln^-/-^ SD or WT HD vs. Nln^-/-^ HD. Statistical analyses were performed using two-way ANOVA with the post-hoc Holm–Sidak test. Data are presented as mean ± SEM. n = 5–8 animals per group.

**Figure 4 ijms-24-15190-f004:**
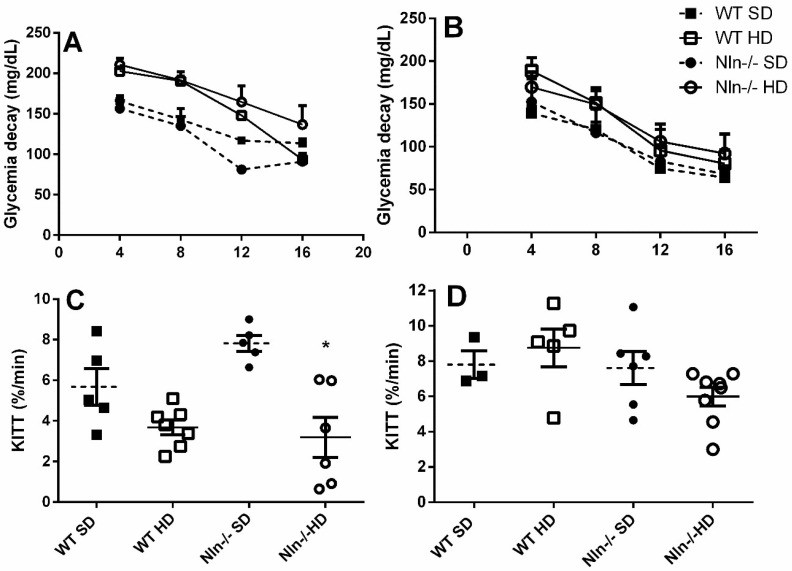
Insulin tolerance test (ITT) was performed in WT and Nln^-/-^, male (**A**,**C**) and female (**B**,**D**) animals fed SD or HD. Intraperitoneal insulin administration was followed by blood glucose measurement after 4, 8, 12 and 16 min (**A**,**B**). Note that only HD-fed male Nln^-/-^ animals showed statistically significant reduced insulin sensitivity (**C**). Nln^-/-^ male animals fed SD have a tendency to have greater insulin sensitivity, which was not seen as statistically significant. (**A**–**D**): *, *p* < 0.05 between WT SD vs. WT HD or Nln^-/-^ SD vs. Nln^-/-^ HD. Statistical analyses were performed using two-way ANOVA with the post-hoc Holm–Sidak test. Data are presented as mean ± SEM. n = 5–8 animals per group.

**Table 1 ijms-24-15190-t001:** Food, water and calorie consumption of male animals.

Consumption:	WT SD	WT HD	Nln^-/-^ SD	Nln^-/-^ HD
Food (g/day/animal)	5.02 ± 0.27	1.87 ± 0.25 *	5.9 ± 0.46	2.57 ± 0.10 *^.#^
Water (mL/day/animal)	1.3 ± 0.15	1.5 ± 0.05	1.5 ± 0.09	1.3 ± 0.16
Calories (Kcal/day/animal)	19.08 ± 0.95	15.98 ± 2.07	22.43 ± 2.24	21.97 ± 1.05

*, represents statistical significance between diets (SD or HD); ^#^, represents statistical significance between genotypes. Data were analyzed using ANOVA two-way and post-test of post-hoc Holm–Sidak. It was considered statistically significant when *p* < 0.05. Data are shown as mean ± SEM. n = 5–8.

**Table 2 ijms-24-15190-t002:** Food, water, and calories consumption of female animals.

Consumption:	WT SD	WT HD	Nln^-/-^ SD	Nln^-/-^ HD
Food (g/day/animal)	3.91 ± 0.59	1.39 ± 0.15 *	4.38 ± 0.20	2.09 ± 0.16 *^. #^
Water (mL/day/animal)	1.6 ± 0.1	2.0 ± 0.2 *	1.7 ± 0.2	1.0 ± 0.1 ^#^
Calories (Kcal/day/animal)	14.3 ± 1.56	11.88 ± 1.20	16.6 ± 0.83	17.86 ± 0.90

*, represents statistical significance between diets (SD or HD); ^#^, represents statistical significance between genotypes. Data were analyzed using ANOVA two-way and post-test of post-hoc Holm–Sidak. It was considered statistically significant when *p* < 0.05. Data are shown as mean ± SEM. n = 5–8.

**Table 3 ijms-24-15190-t003:** The percentage of wet mass of different tissues from WT or Nln^-/-^ corrected by the animals’ body mass (mg/g).

Tissues	WT SD	WT HD	Nln^-/-^ SD	Nln^-/-^ HD
Liver	4.48 ± 0.09	2.95 ± 0.07 *	4.70 ± 0.16	2.90 ± 0.10 *
Inguinal Fat	0.80 ± 0.06	2.24 ± 0.30 *	0.76 ± 0.06	2.98 ± 0.29 *
Gonadal Fat	1.31 ± 0.12	3.67 ± 0.47 *	0.92 ± 0.09	4.37 ± 0.25 *
Retroperitoneal Fat	0.38 ± 0.08	1.34 ± 0.13 *	0.33 ± 0.06	2.36 ± 0.13 *^,#^
Gastrocnemius muscle	1.15 ± 0.02	0.96 ± 0.03 *	1.07 ± 0.04	0.89 ± 0.08*

*, represents statistical significance between diets (SD or HD); ^#^, represents statistical significance between genotypes. Data were analyzed using ANOVA two-way and post-test of post-hoc Holm–Sidak. It was considered statistically significant when *p* < 0.05. Data are shown as mean ± SEM. n = 5–8.

**Table 4 ijms-24-15190-t004:** The percentage of wet mass of different tissues from females WT or Nln^-/-^ corrected by the animals’ body mass (mg/g).

Tissues	WT SD	WT HD	Nln^-/-^ SD	Nln^-/-^ HD
Liver	4.22 ± 0.35	3.45 ± 0.09 *	4.4 ± 0.18	3.32 ± 0.09 *
Inguinal Fat	1.01 ± 0.11	1.92 ± 0.13 *	0.86 ± 0.07	3.26 ± 0.19 *^,#^
Gonadal Fat	1.10 ± 0.17	2.76 ± 0.26 *	0.61 ± 0.11	4.29 ± 0.41 *^,#^
Retroperitoneal Fat	0.39 ± 0.11	1.16 ± 0.11 *	0.27 ± 0.06	2.55 ± 0.23 *^,#^
Gastrocnemius muscle	1.10 ± 0.02	1.05 ± 0.02	1.13 ± 0.08	0.94 ± 0.04 *

*, represents statistical significance between diets (SD or HD); **^#^**, represents statistical significance between genotypes. Data were analyzed using ANOVA two-way and post-test of *post-hoc* Holm–Sidak. It was considered statistically significant when *p* < 0.05. Data are shown as mean ± SEM. n = 5–8.

**Table 5 ijms-24-15190-t005:** Gene expression levels in adipose tissue from male WT and Nln^-/-^ mice analyzed by RT-PCR, compared to constitutively expressed RPL19.

Gene	WT SD	WT HD	Nln^-/-^ SD	Nln^-/-^ HD
**PPAR-alpha**	1.04 ± 0.22	54.23 ± 6.66 *	5.46 ± 0.85 ^#^	47.71 ± 6.54 *
**PPAR-gamma**	1.02 ± 0.05	0.82 ± 0.12	0.51 ± 0.01 ^#^	0.43 ± 0.10 ^#^
**FAS**	1.01 ± 0.04	0.8 ± 0.15	0.98 ± 0.17	1.04 ± 0.17
**LPL**	1.02 ± 0.18	7.64 ± 0.92 *	4.4 ± 1.29 ^#^	7.13 ± 1.54 *
**FABP4**	1.02 ± 0.15	1.84 ± 0.27	1.48 ± 0.29	2.94 ± 0.78 *
**Leptin**	1.02 ± 0.14	16.16 ± 3.83 *	1.54 ± 0.49	46.16 ± 6.89 *^,#^
**CD206**	1.02 ± 0.04	3.63 ± 1.33 *	1.86 ± 0.30	4.07 ± 0.42 *
**CD11C**	1.05 ± 0.26	7.28 ± 2.46 *	1.43 ± 0.6	14.75 ± 6.05 *
**PGC-1 alpha**	1.02 ± 0.07	0.74 ± 0.05	0.8 ± 0.08	0.79 ± 0.20
**F4-80**	1.02 ± 0.07	2.87 ± 1.26 *	1.0 ± 0.14	3.98 ± 0.63 *
**CD36**	1.00 ± 0.08	2.33 ± 0.57	1.88 ± 0.41	2.07 ± 0.48
**ADBR3**	1.02 ± 0.18	0.59 ± 0.08 *	0.48 ± 0.13 ^#^	0.32 ± 0.09 ^#^
**NEP**	1.00 ± 0.01	1.22 ± 0.09	1.37 ± 0.27	1.73 ± 0.27
**POP**	1.00 ± 0.06	1.33 ± 0.30	1.03 ± 0.16	1.33 ± 0.13
**DPP4**	1.00 ± 0.03	1.72 ± 0.27	1.70 ± 0.28	1.98 ± 0.50
**IDE**	1.01 ± 0.09	6.46 ± 0.93 *	4.59 ± 0.97 ^#^	8.33 ± 0.36 *
**ACE1**	1.03 ± 0.21	2.30 ± 0.57 *	2.06 ± 0.30 ^#^	2.89 ± 0.39 *
**β5-Prot**	1.01 ± 0.13	2.72 ± 0.31 *	2.02 ± 0.41 ^#^	4.03 ± 0.40 *^,#^

*, represents statistical significance between diets (SD or HD); **^#^**, represents statistical significance between genotypes. The gene expression levels were normalized with RPL19 as a housekeeping gene. Data were analyzed using ANOVA two-way and post-test of *post-hoc* Holm–Sidak. It was considered statistically significant when *p* < 0.05. Data are shown as mean ± SEM. n = 5–8.

**Table 6 ijms-24-15190-t006:** Gene expression levels in adipose tissue from male WT and Nln^-/-^ mice analyzed by RT-PCR, compared to constitutively expressed cyclophilin A.

Gene	WT SD	WT HD	Nln^-/-^ SD	Nln^-/-^ HD
**PPAR-alpha**	1.24 ± 0.60	29.7 ± 3.71 *	5.28 ± 1.58 ^#^	25.30 ± 5.59 *
**PPAR-gamma**	1.16 ± 0.41	0.86 ± 0.18	0.51 ± 0.11 ^#^	0.39 ± 0.06 ^#^
**FAS**	1.10 ± 0.33	0.43 ± 0.04	0.69 ± 0.06	0.55 ± 0.11
**LPL**	1.09 ± 0.29	23.76 ± 3.41 *	14.45 ± 3.50 ^#^	29.17 ± 6.84 *
**FABP4**	1.03 ± 0.17	1.52 ± 0.28	1.39 ± 0.21	2.25 ± 0.65 *
**Leptin**	1.06 ± 0.23	9.09 ± 2.66 *	2.21 ± 0.91	24.11 ± 4.69 *^,#^
**CD206**	1.04 ± 0.23	2.99 ± 1.26 *	2.03 ± 0.43	3.04 ± 0.05 *
**CD11C**	1.03 ± 0.17	6.18 ± 2.28 *	2.20 ± 1.37	12.63 ± 5.55 *
**PGC-1 alpha**	1.03 ± 0.18	0.58 ± 0.01	1.04 ± 0.23	0.63 ± 0.21
**F4-80**	1.07 ± 0.27	3.67 ± 1.81 *	1.62 ± 0.17	4.87 ± 1.18 *
**CD36**	1.04 ± 0.21	1.83 ± 0.34	1.28 ± 0.38	1.57 ± 0.41
**ADBR3**	1.21 ± 0.55	0.69 ± 0.04 *	0.67 ± 0.07 ^#^	0.34 ± 0.05 ^#^
**NEP**	1.06 ± 0.26	0.99 ± 0.13	1.34 ± 0.19	1.28 ± 0.04
**POP**	1.12 ± 0.35	1.30 ± 0.35	1.23 ± 0.12	1.21 ± 0.15
**DPP4**	1.08 ± 0.29	1.48 ± 0.27	2.16 ± 0.56	1.52 ± 0.19
**IDE**	1.11 ± 0.31	3.53 ± 0.48 *	2.93 ± 0.43 ^#^	4.31 ± 0.36 *
**ACE1**	1.00 ± 0.11	1.76 ± 0.51 *	2.09 ± 0.51 ^#^	2.01 ± 0.02
**β5-Prot**	1.00 ± 0.10	1.82 ± 0.21 *	1.57 ± 0.09 ^#^	2.51 ± 0.07 *^,#^

*, represents statistical significance between diets (SD or HD); **^#^**, represents statistical significance between genotypes. The mRNA levels were normalized with cyclophilin A as a housekeeping gene. Data were analyzed using ANOVA two way and post-test of *post-hoc* Holm–Sidak. It was considered statistically significant when *p* < 0.05. Data as shown as mean ± SEM. n = 5–8.

**Table 7 ijms-24-15190-t007:** Gene expression levels in adipose tissue from female WT and Nln^-/-^ mice analyzed by RT-PCR, compared to constitutively expressed RPL19.

Gene	WT SD	WT HD	Nln^-/-^ SD	Nln^-/-^ HD
**PPAR-alpha**	2.75 ± 0.17	12.31 ± 3.74 *	6.25 ± 1.95 ^#^	16.13 ± 1.17 *
**PPAR-gamma**	0.43 ± 0.10	0.77 ± 0.15	0.63 ± 0.11	0.60 ± 0.06
**FAS**	0.73 ± 0.07	0.36 ± 0.02 *	0.98 ± 0.08	0.72 ± 0.10 ^#^
**LPL**	1.07 ± 0.08	4.19 ± 0.55 *	0.56 ± 0.1	4.86 ± 0.37 *
**FABP4**	0.72 ± 0.04	1.61 ± 0.29 *	0.55 ± 0.14	1.27 ± 0.12 *
**Leptin**	2.91 ± 0.84	8.02 ± 1.29 *	2.94 ± 0.63	25.44 ± 7.42 *^,#^
**CD206**	0.60 ± 0.02	1.98 ± 0.28 *	0.71 ± 0.14	2.28 ± 0.29 *
**CD11C**	0.84 ± 0.15	2.09 ± 0.34 *	0.62 ± 0.13	4.42 ± 0.68 *
**PGC-1 alpha**	0.93 ± 0.12	0.57 ± 0.12	0.46 ± 0.11^#^	0.57 ± 0.07
**F4-80**	0.26 ± 0.03	1.25 ± 0.08 *	0.45 ± 0.07	1.80 ± 0.23 *
**CD36**	0.69 ± 0.05	2.18 ± 0.42 *	0.57 ± 0.14	2.45 ± 0.55 *
**ADBR3**	0.39 ± 0.07	0.91 ± 0.19 *	0.40 ± 0.08	0.36 ± 0.06 ^#^
**NEP**	0.79 ± 0.10	1.14 ± 0.16	0.90 ± 0.11	1.29 ± 0.11
**POP**	0.89 ± 0.16	0.96 ± 0.12	0.96 ± 0.10	1.34 ± 0.13
**DPP4**	0.82 ± 0.21	1.35 ± 0.27 *	0.87 ± 0.12	1.45 ± 0.20 *
**IDE**	4.85 ± 1.14	5.98 ± 0.65	4.23 ± 0.90	6.96 ± 0.93
**ACE1**	0.62 ± 0.09	1.15 ± 0.25 *	0.60 ± 0.10	1.45 ± 0.17 *
**β5-Prot**	1.16 ± 0.11	2.16 ± 0.17	1.92 ± 0.26	3.32 ± 0.44

*, represents statistical significance between diets (SD or HD); **^#^**, represents statistical significance between genotypes. The gene expression levels were normalized with RPL19 as a housekeeping gene. Data were analyzed using ANOVA two-way and post-test of *post-hoc* Holm–Sidak. It was considered statistically significant when *p* < 0.05. Data are shown as mean ± SEM. n = 5–8.

**Table 8 ijms-24-15190-t008:** Gene expression levels in adipose tissue from female WT and Nln^-/-^ mice analyzed by RT-PCR, compared to constitutively expressed cyclophilin A.

Gene	WT SD	WT HD	Nln^-/-^ SD	Nln^-/-^ HD
**PPAR-alpha**	3.29 ± 1.21	9.66 ± 0.38 *	5.67 ± 1.75	17.35 ± 1.48 *
**PPAR-gamma**	0.97 ± 0.29	1.16 ± 0.26	0.66± 0.14	1.04 ± 0.23
**FAS**	0.90 ± 0.05	0.33 ± 0.05 *	1.13 ± 0.34	0.71 ± 0.12 ^#^
**LPL**	11.91 ± 2.98	24.08 ± 2.29 *	3.94 ± 1.87	26.48 ± 6.07 *
**FABP4**	1.30 ± 0.55	1.97 ± 0.34	0.94 ± 0.21	1.53 ± 0.22 *
**Leptin**	2.60 ± 0.71	7.34 ± 0.45 *	2.70 ± 0.59	16.71 ± 3.36 *^,#^
**CD206**	0.97 ± 0.09	2.31 ± 0.25 *	0.81 ± 0.24	3.03 ± 0.63 *
**CD11C**	1.56 ± 0.13	2.61 ± 0.42 *	0.85 ± 0.16	6.06 ± 1.31 *
**PGC-1 alpha**	1.53 ± 0.11	0.98 ± 0.19	0.69 ± 0.16^#^	1.27 ± 0.47
**F4-80**	0.86 ± 0.13	2.05 ± 0.30 *	0.70 ± 0.17	3.09 ± 0.38 *
**CD36**	1.51 ± 0.28	2.66 ± 0.47 *	0.63 ± 0.19	3.35 ± 0.84 *
**ADBR3**	1.18 ± 0.60	1.57 ± 0.31	0.51 ± 0.07	0.61 ± 0.11 ^#^
**NEP**	1.20 ± 0.18	1.33 ± 0.20	1.36 ± 0.17	1.91 ± 0.42
**POP**	1.49 ± 0.38	1.33 ± 0.19	1.48 ± 0.07	2.36 ± 0.67
**DPP4**	0.99 ± 0.19	1.70 ± 0.34 *	1.00 ± 0.09	2.19 ± 0.68 *
**IDE**	4.93 ± 0.85	4.56 ± 0.86	5.60 ± 0.79	7.22 ± 2.47
**ACE1**	0.95 ± 0.06	1.38 ± 0.30 *	0.99 ± 0.01	1.88 ± 0.51 *
**β5-Prot**	2.13 ± 0.47	2.05 ± 0.21	1.49 ± 0.27	3.25 ± 0.72

*, represents statistical significance between diets (SD or HD); **^#^**, represents statistical significance between genotypes. The mRNA levels were normalized with cyclophilin A as a housekeeping gene. Data were analyzed using ANOVA two way and post-test of *post-hoc* Holm–Sidak. It was considered statistically significant when *p* < 0.05. Data as shown as mean ± SEM. n = 5–8.

**Table 9 ijms-24-15190-t009:** Sequences of primers used in RT-qPCR experiments.

Gene	Sequence	Amplicon (bp)	Access Number
Peroxisome proliferator activated receptor gamma (PPAR-gamma)	Fwd: ATCTTAACTGCCGGATCC Rev: CAAACCTGATGGCATTGTGAG	102	NM_001127330.2
Peroxisome proliferator activated receptor alpha (PPAR-alpha)	Fwd: TGCAATTCGCTTTGGAA Rev: CTTGCCCAGAGATTTGAGGT	118	NM_011144.6
Fatty acid synthase (FAS)	Fwd: GATTCGGTGTCTGCTGTC Rev: CATGCTTTAGCACCTGCTGT	95	NM_007988.3
Lipoprotein lipase (LPL)	Fwd: GTCTGGCCACTGGACAAA Rev: CCCACTTTCAAACACCCAAA	122	NM_008509.2
Leptin	Fwd: CCAGGATGACACCAAAACCCT Rev: TGAAGTCCAAGCCAGTGACC	107	NM_008493.3
CD36 molecule (CD36)	Fwd: GATTGGTTGAGACCCCG Rev: GCTCCACACATTTCAGAAGGC	174	NM_001159558.1
Peroxisome proliferator-activated receptor gamma coactivator 1-alpha (PGC 1 alpha)	Fwd: AAGGGCCAAACAGAGAG Rev: AGTAAATCACACGGCGCTCTT	63	NM_008904.3
Mannose receptor (CD206)	Fwd: TGTGTTCAGCTATTGGACGC Rev: CGGAATTTCTGGGATTCAGCTTC	133	NM_008625.2
Integrin alpha X (CD11C)	Fwd: CTGGATAGCCTTTCTTCTGCTG Rev: GCACACTGTGTCCGAACTCA	113	NM_021334.3
Adhesion G protein-coupled receptor E1 (F4/80)	Fwd: AACATGCAACCTGCCACAAC Rev: TTCACAGGATTCGTCCAGGC	110	NM_010130.4
Fatty acid-binding protein 4 (FABP4)	Fwd: CGCAGACGACAGGAAGGT Rev: TTCCATCCCACTTCTGCAC	77	NM_024406.3
Ribosomal protein L19 (RPL19)	Fwd: CAATGCCAACTCCCGTCA Rev: GTGTTTTTCCGGCAACGAG	102	NM_009078.2
Adrenergic receptor, beta 3 (ADBR3)	Fwd: ACCCTGATGATCGACATGTTCC Rev: GCCATAGTGAGGAGACAGGG	129	NM_013462.3
Neprilysin (NEP)	Fwd: CCTGAACTTTGCCCAGGTGT Rev: GCGGCAATGAAAGGCATCTG	148	NM_001289462.1
Angiotensin I converting enzyme (ACE1)	Fwd: ACCCTAGGACCTGCCAATCT Rev: CGTGAGGAAGCCAGGATGTT	164	NM_207624.5
Prolyl oligopeptidase (POP)	Fwd: GGGTGCTCCGACACTAAACA Rev: GACGGGTACTGGATGTCGTC	98	NM_011156.3
Insulin degrading enzyme (IDE)	Fwd: GTCCATGTTCTTGCCAGGGA Rev: TTCACGAGGGGAAACAGTGG	161	NM_031156.3
Dipeptidyl peptidase 4 (DPP4)	Fwd: GACGGCAGAGGAAGTGGTT Rev: CGCTTGCTATCCACAAATCCC	134	NM_010074.3
Proteasome subunit beta 5 (Prot-β5)	Fwd: CCAAACTGCTCGCTAACATGG Rev: GTTCCCCTCGCTGTCTACG	119	NM_011186.1
Ribosomal protein L19 (RPL19)	Fwd: CAATGCCAACTCCCGTCA Rev: GTGTTTTTCCGGCAACGAG	102	NM_009078.2
Cyclophilin A	Fwd: TATCTGCACTGCCAAGACTGAGT Rev: CTTCTTGCTGGTCTTGCCATTCC	127	NM_008907.2

## Data Availability

The data presented in this study are available on request from the corresponding author.

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
