# Peer review of "Neurolysin Knockout Mice in a Diet-Induced Obesity Model"

_ijms, 2023, doi:10.3390/ijms242015190_

Round 1

Reviewer 1 Report

The paper by Caprioli et al. concerns the influence of the neurotensin-degrading enzyme neurolysin in a diet-induced mice model of obesity. The paper is descriptive and reports on a rater complete characterization of several paradigms, namely food intake and body. Authors report on a higher food intake and gain in body mass elicited by neurolysin gene depletion. The experiments are well-controlled and, although the mechanistic aspects underlying these observations remain to be delineated, the paper is of interest and could be published. I have only a few comments concerning the bibliographic citations.

In the Keywords list, Neurolysin should be added

In the introduction

At 29-30. Add Vincent et al. J. Neurosci. 1996.

25,35,36 Add Barelli et al. Neurochemistry international 1993

26,27,43,44 Checler F et al.  J. Biol. Chem  1986

In the discussion, authors could discuss the putative complementarity/redundancy of neurolysin and Thimet oligoopeptidase, which display rather similar although not completely identical, specificity for small peptides.

Authors could also discuss Neurolysisn-mediated phenotype with respect with its dual astrocytic-neuronal localization.

Author Response

Reviewer #1

Dear reviewer,

We kindly appreciate your time and attention to review our manuscript, and would like to thank you very much for that. Please, find below (and/or in the attachment) the point-to-point answers to your comments. 

Sincerely,

Emer S. Ferro, PhD

1) In the Keywords list, Neurolysin should be added

Answer: We have added Neurolysin to the key words list

 2) In the introduction

At 29-30. Add Vincent et al. J. Neurosci. 1996.

25,35,36 Add Barelli et al. Neurochemistry international 1993

26,27,43,44 Checler F et al.  J. Biol. Chem  1986

Answer: We have added the references above to the Introduction. However, we have not been able to identify the reference from Barelli et al, 1993 at Neurochemistry International. Therefore, we have added the article about “Purification, physico-chemical characteristics and differential specificity towards opiates, tachykinins and neurotensin-related peptides”, published by Barelli et al, in the European Journal of Biochemistry in 1993. We hope this reference is appropriate.

3) In the discussion, authors could discuss the putative complementarity/redundancy of neurolysin and Thimet oligoopeptidase, which display rather similar although not completely identical, specificity for small peptides.

Answer: In the Discussion, we have discussed the added the following paragraph:

“Despite the biochemical similarities of Nln and THOP1 among most substrates, neurotensin (pyr-Glu-Leu-Tyr-Glu-Asn-Lys-Pro-Arg-Arg-Pro-Tyr-Ile-Leu-OH) is cleaved by Nln at the Pro10-Tyr11 bond, while THOP1 cleaves the Arg8-Arg9 bond. Moreover, Nln and THOP1 can also be distinguished by their sensitivity to carboalkyls, phosphinic and Pro-Ile inhibitors[42-44] and their drastic distinct sensitivity to thiol compounds[45-47].”

Reviewer 2 Report

I’ve carefully read this paper about the impact of high caloric diet on metabolic features of Neurolysin knock-out mice. The results are interesting but mainly descriptive. Analysis of Neurolysin involvement is limited due to the use of general knock-out mice, but these results could be interesting for scientist working in the field. Nevertheless, I have some major concerns which will be I hope fixed by the authors.

-          Introduction is badly organized without links between the different parts. The authors must reorganize the introduction and clearly highlighted the interest to study Neurolysin in a context of obesity.

-          In part 2.1. age of mice and housing temperature must be specified.

-          One housekeeping gene is poor rigorous for qPCR. Authors may envisage to add one more hkg.

-          Insulin and glucose quantity must be noted as g/kg of body weight. To note for authors, it would be better for GTT and especially ITT to determine glucose and insulin quantity using lean mass as reference instead of fat mass. This point is more and more recommended by all scientists working in metabolism field. Indeed, as muscle and liver are primary target, performing GTT and ITT with dose of glucose and insulin taking account mainly fat mass can lead to overinterpretation or misinterpretation. Thus, the conclusions of this work concerning ITT and GTT must be interpreted with caution.

-          Genotyping experiments are not interesting and must be excluded of the result part.

-          How authors explain that their mice (especially female) under HD did not display more body weight increase (figure 2). It is largely different from the previous publications using this diet and ask question about the quality of this study.

-          I’m convinced that calories in Table 2 have not been correctly calculated as they don’t take account of the condensed milk. Otherwise, it will be an experiment of High caloric diet where mice fed less calories than mice feeding standard diet.

-          In the discussion the authors argue that their results support a leptin resistance phenotype in Neurolysin knock out mice. I’m not convinced that the results displayed in this paper support this point. Indeed, we know that leptin expression, and ultimately secretion, increases linearly with increasing body fat, which is the case here. Moreover, we can expect a stronger increase in food intake and body weight gain in case of leptin resistance, which is not the case in neurolysin knock-out mice. Finally, analysis of leptin expression at mRNA level is not sufficient to draw any conclusions.

-          Finally, authors analysed mouse metabolism at classic housing temperature (I mind) using a model of neurolysin ko and High caloric diet. Knowing that neurolysin has a role in central nervous system, it is difficult to appreciate the results displayed in this paper without characterization of brown fat, physical activity, and feeding behaviour; three aspects of mouse physiological metabolism known to be modulated by the CNS, and which could explain the results obtained here. These points should be analysed, and discussed taking account of the results which will be available.

-          The results presented in this work should be further discussed in relation to the authors' previously published paper characterizing the phenotype of Neurolysin ko mice fed a standard diet.

Author Response

Reviewer #2

Dear reviewer,

We kindly appreciate your time and attention to review our manuscript, and would like to thank you very much for that. Please, find below (and attached) the point-to-point answers to your comments.

1) -          Introduction is badly organized without links between the different parts. The authors must reorganize the introduction and clearly highlighted the interest to study Neurolysin in a context of obesity.

Answer: We have reorganized the Introduction, and have added a new paragraph (please, see below) to justify the investigation of Nln knockout animals under a diet-induced obesity model.

“Nln and thimet oligopeptidase (EC3.4.24.15; THOP1) are closely related peptidases [18,33]. Recently, THOP1 was shown to play a key role energy metabolism expenditure, and THOP1 knockout mice (THOP1-/-) were shown resistant to diet-induced obesity[34]. Here, Nln-/- and WT animals, male and female, were challenged by a diet-induced obesity model. Body mass and several other metabolic parameters were evaluated to advance the understanding of the biological relevance of Nln in energy metabolism.”

 2) -          In part 2.1. age of mice and housing temperature must be specified.

Answer: We have added the mice age at the first paragraph of section 2.1, as following: “Male and female wild type C57BL6/6N mice (WT), eight-week-old, were obtained...”. We have also added the housing temperature, as following: “The mice were maintained in individually ventilated cages (Ventilife, ALESCO, SP, Brazil) at 22°C, under...”

3) -          One housekeeping gene is poor rigorous for qPCR. Authors may envisage to add one more hkg.

Answer: In fact we have used cyclophilin A and beta-actin as internal relative controls, in addition to RPL19. However, results were similar and we ended showing only RPL19. We have now mentioned that on the text, as following:

“and expressed as relative values using the “2ddCt” method, as previously described[34]. In addition to RPL19, cyclophilin A (sense: TATCTGCACTGCCAAGACTGAGT; anti-sense: CTTCTTGCTGGTCTTGCCATTCC) and beta actin (sense: AAGATTTGGCACCACACTTTCTACA; anti-sense: CGGTGAGCAGCACAGGGT) were genes evaluated as internal relative controls, producing similar results to that of RPL19 (data not shown). Target gene expression levels of WT animals were compared to Nln-/-,…”

4) Insulin and glucose quantity must be noted as g/kg of body weight. To note for authors, it would be better for GTT and especially ITT to determine glucose and insulin quantity using lean mass as reference instead of fat mass. This point is more and more recommended by all scientists working in metabolism field. Indeed, as muscle and liver are primary target, performing GTT and ITT with dose of glucose and insulin taking account mainly fat mass can lead to overinterpretation or misinterpretation. Thus, the conclusions of this work concerning ITT and GTT must be interpreted with caution.

Answer: We thank the reviewer for these comments. We are now aware that our present results need to be interpreted with caution.

5) Genotyping experiments are not interesting and must be excluded of the result part.

Answer: we have removed these results from the manuscript accordingly.

6)          How authors explain that their mice (especially female) under HD did not display more body weight increase (figure 2). It is largely different from the previous publications using this diet and ask question about the quality of this study.

Answer: The present protocol of diet-induced obesity was conducted for only 4 weeks. As can be seen in the Figure 1, both male and female fed HD gained more weight than their respective controls fed SD. Therefore, the reviewer must take in consideration the short period of time that these experiments were conducted. Indeed, in our previous experience using a diet-induced obesity without condensed milk, it was necessary 12 weeks until the WT and thimet oligopeptidase knockout mice could start to show differences on their body mass. In that previous study we have fed the animals for 24 weeks, and the differences in the body weigh of animals fed SD vs HD was more evident as expected. However, in the present study there were differences in the body mass of the animals fed SD vs HD, as shown by the statistical analysis.

7) I’m convinced that calories in Table 2 have not been correctly calculated as they don’t take account of the condensed milk. Otherwise, it will be an experiment of High caloric diet where mice fed less calories than mice feeding standard diet.

Answer: We thank the reviewer for these comments. However, we have correctly calculated the calories, considering the addition of condensed milk. Note that these mice gained all the weight in only 4 weeks of diet-induced obesity. In our previous experience using a diet-induced obesity without condensed milk it took 12 weeks until wild type C57BL6/N mice start to gain weight distinctively from the thimet oligopeptidase knockout mice. Here, in only 4 weeks of diet we were able to see the differences between Nln-/- and wild type mice. The fact that mice under HD ingested less calories compared to wild type mice, despite the fact that these mice gained more weight, is an indication that condensed milk drives energy to fat accumulation more than kinetic movement energy expenditure.

8) In the discussion the authors argue that their results support a leptin resistance phenotype in Neurolysin knock out mice. I’m not convinced that the results displayed in this paper support this point. Indeed, we know that leptin expression, and ultimately secretion, increases linearly with increasing body fat, which is the case here. Moreover, we can expect a stronger increase in food intake and body weight gain in case of leptin resistance, which is not the case in neurolysin knock-out mice. Finally, analysis of leptin expression at mRNA level is not sufficient to draw any conclusions.

Answer: Thank you for your comments. We have indeed observed a linear increase of leptin expression parallel with body weight gain, which was greater in Nln-/- mice. We agree with the reviewer that further experiments should be necessary to investigate this possibility.

9)  Finally, authors analyzed mouse metabolism at classic housing temperature (I mind) using a model of neurolysin ko and High caloric diet. Knowing that neurolysin has a role in central nervous system, it is difficult to appreciate the results displayed in this paper without characterization of brown fat, physical activity, and feeding behaviour; three aspects of mouse physiological metabolism known to be modulated by the CNS, and which could explain the results obtained here. These points should be analysed, and discussed taking account of the results which will be available.

Answer: We really appreciate your comments, and definitively will take these physiological aspects in consideration on our further investigations.

10) The results presented in this work should be further discussed in relation to the authors' previously published paper characterizing the phenotype of Neurolysin ko mice fed a standard diet.

Answer: We thank the reviewer for these comments. In the introduction, we have reviewed the previous characterized phenotype of Nln-/- mice fed SD. However, we have now reorganized the discussion to make it clear, accordingly to reviewers’ comments.

Round 2

Reviewer 2 Report

Dear authors,

I cannot take your very limited responses into account in its current form, as you do not provide any new results or explicit answers to solve the main problems reported. I read exactly the first version with only a few adjustments. Below are my comments on several of these points.

3) Authors have added this sentence: “In addition to RPL19, cyclophilin A (sense: TATCTGCACTGCCAAGACTGAGT; anti-sense: CTTCTTGCTGGTCTTGCCATTCC) 181 and beta actin (sense: AAGATTTGGCACCACACTTTCTACA; anti-sense: CGGTGAG-182 CAGCACAGGGT) were genes evaluated as internal relative controls, producing similar results to that of RPL19 (data not shown).” Thus, they authors must be able to display a new analysis taking account of these two new hkg in the same experiment.

6)   The authors’ response is not in line with their manuscript. As indicated in legend of new figure 1 :” Body mass gain in male (A) or female (B) mice, WT or Nln-/-, fed standard (SD) or hyper-caloric (HD) diet. Variation of body mass (mass delta) from male (C) or female (D) along the eighth weeks receiving SD or HD (final body mass subtracted from initial body mass).”  As well as in the main text :” After eight weeks on diet, all animals receiving HD, as expected, had greater body mass gain compared to SD fed animals, regardless of genotype or gender (Fig. 1).” The authors must clearly fix this point and indicated clearly in the text and figures the exact protocol used in the present work.

7) If calorie intake results did not consider the calorie intake due to condensed milk, there is no sense to display it as it is not informative. The authors can leave these results by precising the fact that the calories are only calculated on a part of the diet but they authors can’t use it as a result sustaining their discussion in the present form.

8) The authors agree that increased leptin correlates with increased body weight but have not modified their discussion accordingly. Again, the slight increase in leptin mRNA cannot be used to propose leptin resistance, at least as stated in the discussion.

9)  Definitively, the authors do not take account of this comment on metabolism analysis and brown adipose tissue. No additional results as well as discussion have been added in the new manuscript.

Author Response

Reviewer #2

Dear reviewer,

I apologize enormously for misunderstanding your main questions. We definitively appreciate your time and quick response to our previous revised manuscript. Please, find below the point-to-point answers to your new comments.

3) Authors have added this sentence: “In addition to RPL19, cyclophilin A (sense: TATCTGCACTGCCAAGACTGAGT; anti-sense: CTTCTTGCTGGTCTTGCCATTCC) 181 and beta actin (sense: AAGATTTGGCACCACACTTTCTACA; anti-sense: CGGTGAG-182 CAGCACAGGGT) were genes evaluated as internal relative controls, producing similar results to that of RPL19 (data not shown).” Thus, they authors must be able to display a new analysis taking account of these two new hkg in the same experiment.

Answer: We apologize very much for not been completely assertive on our first responses to the reviewer. We have now added the Table 8 and 9 to the manuscript showing the results from relative gene expression, using cyclophilin as the housekeeping gene. The use of these two housekeeping genes has been previously suggested for sex-unbiased diet-induced obesity mice models (PMC7732482).  On the other hand, we decided not to show the data obtained using the beta-actin as the relative housekeeping gene, because previous reports have suggested beta-actin to be unstable during the process of diet-induced obesity in mice (PMC7732482). It was important to mention that leptin gene expression increased in WT and Nln-/- animals fed HD despite the housekeeping gene used; although, a smaller increase on leptin gene expression was observed considering the beta-actin as the housekeeping gene.

We are showing to the reviewer only, the data obtained using beta-actin as the housekeeping gene below.

Table 1. Gene expression levels in adipose tissue from male WT and Nln-/- mice analyzed by RT-PCR.

Gene

WT SD

WT HD

Nln-/- SD

Nln-/- HD

PPAR-alpha

1.31  + 0.70

10.98 + 2,85*

2.22  + 0.64

10.24  + 3.25*

PPAR-gamma

1.07  + 0.24

0.29  + 0.07*

0.32  + 0.10#

0.14  + 0.01#

FAS

1.12 + 0.04

0.50  + 0.03*

0.41  + 0.05#

0.21  + 0.03*#

LPL

1.02  + 0.16

1.90  + 0.03

2.23  + 0.43

2.60  + 0.05

FABP4

1.01 + 0.11

0.52 + 0.01

0.90 + 0.25

0.93 + 0.29

Leptin

1.03 + 0.18

2.99 + 0.38*

1.19 + 0.43

9.17 + 1.14*#

CD206

1.00 + 0.07

0.94 + 0.24

1.14 + 0.10

1.19 + 0.12

CD11C

1.03 + 0.20

1.98  + 0.05*

1.08  + 0.05

4.73  + 1.80*

PGC-1 alpha

1.0 + 0.07

0.21  + 0.03*

0.69  + 0.26

0.26  + 0.11

F4-80

1.02 + 0.15

1.01 + 0.23

0.67 + 0.07

1.36 + 0.03*

CD36

1.00  + 0.08

0.63  + 0.114

0.97 + 0.15

0.58 + 0.09

ADBR3

1.11 + 0.31

0.21 + 0.03*

0.37 + 0.07#

0.11 + 0.02*#

NEP

1.00 + 0.09

0.34 + 0.03

0.88 + 0.25

0.50 + 0.04

POP

1.03 + 0.18

0.41 + 0.04

0.73 + 0.17

0.44 + 0.04

DPP4

1.03 + 0.18

0.50 + 0.01

1.24 + 0.25

0.60 + 0.11

IDE

1.01 + 0.33

1.22 + 0.07

1.70 + 0.22

1.68 + 0.21*

ECA1

1.01 + 0.11

0.58 + 0.07

1.16 + 0.12

0.79 + 0.09

b5-Prot

1.00 + 0.05

0.68 + 0.03*

1.04 + 0.19

1.07 + 0.14

*, represents statistical significance between diets (SD or HD); #, represents statistical significance between genotypes. The mRNA levels were normalized with beta-actin as a housekeeping gene. Data were analyzed using ANOVA two way and post-test of post-hoc Holm-Sidak. It was considered statistically significant when p < 0.05. Data as shown as Mean ± SEM. n= 5-8.

Table 2. Gene expression levels in adipose tissue from female WT and Nln-/- mice analyzed by RT-PCR.

Gene

WT SD

WT HD

Nln-/- SD

Nln-/- HD

PPAR-alpha

1.22 + 0.12

4.94 + 0.83*

4.21 + 1.26#

6.50 + 1.32*

PPAR-gamma

0.54 + 0.26

0.48 + 0.08

0.45+ 0.12

0.33 + 0.05

FAS

0.39 + 0.09

0.17 + 0.03*

0.74 + 0.30

0.19 + 0.03*

LPL

0.86 + 0.36

1.80 + 0.23*

1.09 + 0.17

1.06 + 0.13

FABP4

0.50 + 0.21

0.83 + 0.09

0.69 + 0.16

0.56 + 0.03

Leptin

1.01 + 0.01

4.23 + 1.11*

1.50 + 0.24

5.82 + 1.14*

CD206

0.39 + 0.14

1.03 + 0.13*

0.58 + 0.19

0.86 + 0.11

CD11C

0.60 + 0.16

1.10 + 0.10

0.58 + 0.15

1.66 + 0.14*

PGC-1 alpha

0.58 + 0.13

0.46 + 0.10

0.47 + 0.13

0.26 + 0.05

F4-80

0.22 + 0.03

0.63 + 0.05*

0.34 + 0.11

0.70 + 0.06*

CD36

0.58 + 0.18

1.11 + 0.16*

0.47 + 0.15

0.97 + 0.19*

ADBR3

0.36 + 0.14

0.61 + 0.11

0.33 + 0.04

0.19 + 0.04#

NEP

0.47 + 0.15

0.55 + 0.05

0.96 + 0.06

0.48 + 0.04

POP

0.54 + 0.17

0.57 + 0.04

1.00 + 0.12

0.56 + 0.06

DPP4

0.38 + 0.11

0.71 + 0.10*

0.72 + 0.01

0.59 + 0.08

IDE

1.74 + 0.22

1.98 + 0.23

3.14 + 0.29

1.71 + 0.29

ECA1

0.35 + 0.08

0.58 + 0.08

0.73 + 0.05#

0.51 + 0.06*

b5-Prot

0.82 + 0.14

1.00 + 0.07

1.17 + 0.27

0.98 + 0.08

*,represents statistical significance between diets (SD or HD); #, represents statistical significance between genotypes. The mRNA levels were normalized with beta-actin as a housekeeping gene. Data were analyzed using ANOVA two way and post-test of post-hoc Holm-Sidak. It was considered statistically significant when p < 0.05. Data as shown as Mean ± SEM. n= 5-8.

Also, for the reviewer only, we have compared the relative gene expression of cyclophilin to RPL19. These data suggested that the numerical extent of variation on the expression specific genes, could be due to differential variation of cyclophilin/RPL19 ratios.

Males

Gene

WT SD

WT HD

Nln-/- SD

Nln-/- HD

Cyclophilin A/RPL19

1,09 + 0.34

1.84 + 0.15

1.45 + 0.26

1.98 + 0.23

Females

Gene

WT SD

WT HD

Nln-/- SD

Nln-/- HD

Cyclophilin A/RPL19

1,21 + 0.22

1.46 + 0.16

1.05 + 0.03

1.23 + 0.27

 6)   The authors’ response is not in line with their manuscript. As indicated in legend of new figure 1 :” Body mass gain in male (A) or female (B) mice, WT or Nln-/-, fed standard (SD) or hyper-caloric (HD) diet. Variation of body mass (mass delta) from male (C) or female (D) along the eighth weeks receiving SD or HD (final body mass subtracted from initial body mass).”  As well as in the main text :” After eight weeks on diet, all animals receiving HD, as expected, had greater body mass gain compared to SD fed animals, regardless of genotype or gender (Fig. 1).”

The authors must clearly fix this point and indicated clearly in the text and figures the exact protocol used in the present work.

Answer: We would like to thank the reviewer for these comments. We have improved the Methods and Figure 1 legend explains these experiments and data in further details.

7) If calorie intake results did not consider the calorie intake due to condensed milk, there is no sense to display it as it is not informative. The authors can leave these results by precising the fact that the calories are only calculated on a part of the diet but they authors can’t use it as a result sustaining their discussion in the present form.

Answer: Again, I`m very sorry for not having corrected this mistake from Table 2 and Table 3. I have now fixed both Table 2 and Table 3, which now clearly shows that calories consumption was not statistically significantly different among the groups. 

8) The authors agree that increased leptin correlates with increased body weight but have not modified their discussion accordingly. Again, the slight increase in leptin mRNA cannot be used to propose leptin resistance, at least as stated in the discussion.

Answer: We totally agree that further investigation is needed to understand the possible leptin-resistance of Nln-/- animals.  We have added a new paragraph emphasizing the role of leptin in obesity and the need of further experiments, as following:

“Leptin, the product of the obese (ob) gene[80], is a key adipocyte-secreted hormone for energy balance, and is involved in obesity development[81-83]. Leptin circulates in blood and acts on the brain to regulate food intake and energy expenditure[83]. The body weight-reducing effects of leptin were not seen augmented by enhanced thermogenesis, suggesting that leptin has effects on body temperature regulation, by opposing torpor bouts and by shifting thermoregulatory thresholds; the central pathways behind these effects are largely unexplored[84]. Leptin-deficient ob/ob mice have fully functional brown adipose tissue, and leptin treatment could not increase thermogenesis both in WT and leptin ob/ob mice[85]. Therefore, leptin in parallel to the effects of other cytokines alters centrally regulated thresholds of thermoregulatory mechanisms[85]. Future experiments investigating the brown adipose tissue metabolism in Nln-/- and WT animals, fed either SD or HD, should shed light on the possible contribution of Nln and InPeps to non-shivering thermoregulation[66,86-90].

Gene expression can be correlated with increased confidence to protein expression[91], and strong linear correlations were previously observed between plasma leptin, leptin secretion, and leptin mRNA levels[92]. Therefore, the relative abundance of several mRNAs was evaluated herein to gain molecular insights into the diet-induced obese phenotype observed for Nln-/- animals. The increase in the leptin hormone gene expression could be one of the most relevant for the obese phenotype of Nln-/- animals. Thus, while leptin gene expression increased approximately sixteen times in male WT animals fed HD, it increased approximately forty-six times in male Nln-/- animals, compared to their respective control animals fed SD. In WT female animals fed HD leptin gene expression increased eight times, while in female Nln-/- animals fed HD leptin gene expression increased approximately twenty-five times compared to control animals fed SD. Leptin levels increase when fat mass increases to suppress appetite until weight is lost[83]. However, leptin resistance can occur to increase the predisposition of individuals to diet-induced obesity, which in turn contributes to a further increase in leptin levels and aggravation of existing leptin resistance in a vicious cycle[93]. Several mechanisms are related to leptin resistance, including 1) structural changes to the leptin hormone itself, 2) its transport across the blood-brain barrier, 3) the malfunctioning of the leptin-receptor and signal transduction through the JAK–STAT signaling pathway[73]. Neurotensin, which was the first neuropeptide characterized as a substrate for Nln[2,3,22,23], has several functions, including the control of food intake and energy balance, possibly through the regulation of gut lipid absorption and fat homeostasis[24]. Neurotensin is also one of the peptides that mediate the hypothalamic action of leptin on feeding[94,95]. Unbalanced neurotensin metabolism in Nln-/- fed HD could significantly impact the leptin-sensitive neural circuitry (i.e. causing leptin resistance), considering that neurotensin is a well-known substrate of Nln[1,96]. Further investigations need to be conducted to determine possible changes in hypothalamic neurotensin levels, both in WT and Nln-/- animals fed either a SD or a HD. In addition, leptin resistance in Nln-/- and WT animals fed HD under the present diet-induced obesity model, with sweetened condensed milk and multivitamin complex addition, also need to be investigated.

9)  Definitively, the authors do not take account of this comment on metabolism analysis and brown adipose tissue. No additional results as well as discussion have been added in the new manuscript.

Answer: Again, I`m very sorry for not appropriately answered the reviewer on her/his previous comments. We have reorganized the manuscript in the Introduction, Methods and Discussion. We have also added the additional house-keeping genes experiments in Supplemental Tables 1 and 2. Unfortunately, we have not investigated the brown adipose tissue of the animals used herein. We intend to include the brown adipose tissue analyses in our future projects. The short ten days period of time that we had to reply to your first comments, were not enough to perform any new experiments. We maintain our colony with a very low number of animals due to costs, and to mate these animals to increase their colonies take a considerable time.

The new Discussion about metabolic aspects is shown below.

“Despite their genotype or gender, animals fed HD gained more body mass while ingesting smaller amounts of food, compared to animals that were fed SD; similar results were observed in a previous study using a similar diet-induced obesity model with sweetened condensed milk and multivitamin complex addition[69]. It has been previously shown that substrate utilization and thermogenesis significantly change following ingestion of different types of carbohydrates, in young healthy lean male volunteers[74]. Therefore, the carbohydrate composition of the diet should be expected to have critical outcomes for energy and macronutrient balance in rodents as well. Indeed, distinct proteins, carbohydrates, and lipids have different metabolic roles in energy homeostasis, and because of that, it has been suggested that diets of similar overall energy content, but with different macronutrient distribution, distinctively modify the appetite, metabolism, and thermogenesis[75]. Ingestion of sweetened condensed milk that contains 53% sucrose and causes a rapid increase in plasma insulin levels, induces energy storage as fat in the adipose tissue, which is directly associated with the development of obesity[76,77]. Moreover, micronutrients may also exacerbate obesity-related metabolic consequences, impairing insulin signaling and dysregulating glycemic control; vitamins A, B1, B6, and B12, as well as selenium and zinc, have been reported to be low in the plasma of obese individuals[78,79]. Nln-/- mice were shown herein to be even more sensitive than WT mice to obesity, induced by a HD. That could be occurring through distinctive InPeps and neuropeptides, both in central and peripherical tissues, that differentially modulate the complex macromolecular interaction network responsible for gene expression and protein-protein interactions controlling energy balance and storage[47-50,72]. In fact, the peptidome of Nln-/- mice also have been previously determined, showing alteration in the relative concentration of InPeps in soleus and gastrocnemius muscles, liver, and epididymal adipose tissue[16], and also of neuropeptides such as Met-enkephalin and octapeptide in the brain[17]. Previously, THOP1-/- male and female mice were shown to gain less body weight than WT control mice, both fed a similar HD[28]. Specific InPeps identified in the adipose tissue of HD-fed THOP1-/- mice were suggested to participate in the observed obesity-resistance phenotype[28]. The molecular mechanism was suggested to involve the physical interaction between InPeps and specific microRNAs, which would lead to an alteration in controlling the expression of genes involved in energy metabolism[28]. Therefore, a differential profile of InPeps and neuropeptides in Nln-/- could be driven its distinctive metabolic routes during the diet-induced obesity model used herein, which should be relevant to the different phenotype of Nln-/- animals characterized herein.”

Round 3

Reviewer 2 Report

I'm disappointed by the inconsistency of the answers and the fact that the authors were not alerted and have no explanation for the low mass gain under HFD at least for the control group. Moreover, I'm convinced that the use of this very particular diet leads to huge biases in the analysis of metabolic parameters.
Nevertheless, these are just my personal convictions and I leave it to future readers to appreciate this work.

Author Response

Dear Reviewer,  I'm really very sorry you were disappointed by our responses to your comments. We honestly did our best trying to address all your comments. We truly believe we have did our best to extensively fix the manuscript, from the Abstract to the Discussion. Regarding the fact that WT mice under HD didn't get too much weight as the reviewer expected, we have mentioned that these animals were fed a HD for only seven weeks. This can be considered a short period of time to have a statistically significant weight gain of the Nln-/- animals compared to WT controls, both fed a HD. In our previous experienceworking on the C57BL6/N genetic background (Biomolecules. 2020 Feb 17;10(2):321. doi: 10.3390/biom10020321), without adding the condensed milk and vitamins, at least 12 weeks of HD was necessary to start inducing body weight gain differences on similar WT animals. Therefore, along the manuscript we have stressed the fact that we have used a combination of high fat/high sugar diet-induced obesity model, and fed the animals with that for only seven weeks.  I completely agree with you that our diet-induced obesity model "leads to huge biases in the analysis of metabolic parameters". However, I`m also completely convinced that any diet-induced obesity model is bias. Indeed, the weight differences observed in our present study were sufficient to characterize the different gain of mass between WT and Nln-/- animals fed a HD, which was particularly evident on the adipose tissues analyzed. Even considering that the animals were fed the high fat/high sugar diet for only seven weeks.

Despite the above comments, we have added a new sentence on the manuscript Discussion, second paragraph, addressing the reviewer`s questioning, as following: "The relatively short period of time these animals were fed either with SD or HD, may have impacted the low body mass gain of WT animals HD-fed, compared to those WT animals SD-fed. C57BL/6N WT mouse strain employed herein could have also impacted the low body mass increment observed, as previously reported[77-79]." 

One again, thank you so much for your time, attention, critical reading and comments to our work. It was greatly appreciated.    

Sincerely,

Emer S. Ferro, PhD